# Late Intracerebral Hemorrhage After Successful Endovascular Closure of a Carotid-Cavernous Fistula: A Case Report and Updated Review

**DOI:** 10.3390/reports8040234

**Published:** 2025-11-13

**Authors:** Karol Uscamaita, Marta García Pla, Mikel Terceño, Adrià Arboix, Yolanda Silva

**Affiliations:** 1Neurology Service, Sleep Disorders Unit, Hospital Universitari Sagrat Cor-Grupo Quirónsalud, Universitat de Barcelona, 08034 Barcelona, Spain; karol.uscamaita@quironsalud.es; 2Medicine Department, Universitat de Barcelona, 08036 Barcelona, Spain; 3Emergency Department, Hospital Universitari Sagrat Cor-Grupo Quirónsalud, Universitat de Barcelona, 08034 Barcelona, Spain; martagarcia.bcn.ics@gencat.cat; 4Cerebrovascular Pathology Research Group, Hospital Universitari Dr. Josep Trueta de Girona, Girona Biomedical Research Institute (IDIBGI), 17190 Girona, Spainysilva.girona.ics@gencat.cat (Y.S.); 5Stroke Unit, Department of Neurology, Hospital Universitari Sagrat Cor-Grupo Quirónsalud, Universitat de Barcelona, 17007 Barcelona, Spain

**Keywords:** carotid-cavernous sinus fistula, endovascular procedures, intracerebral hemorrhage

## Abstract

**Background and Clinical Significance**: Intracerebral hemorrhage (ICH) is a very rare complication following endovascular closure of direct carotid-cavernous fistulas (CCFs). When reported, ICH typically appears within the first 48 h after CCF closure. We performed an extensive literature review, starting from the case of a 48-year-old patient presenting with an intracerebral hemorrhage after CCF closure. **Case Presentation**: A 48-year-old woman with arterial hypertension developed an intracerebral hemorrhage in the right frontal lobe 12 days after successful closure of a traumatic CCF. The patient exhibited acute neurological deterioration in a previously hypoperfused territory. A narrative review identifies the classical molecular theory of hemodynamic dysregulation, known as Normal Perfusion Pressure Breakthrough (NPPB), as the principal pathophysiological mechanism. Other mechanisms such as oxidative stress, microglial activation, blood–brain barrier disruption, metalloproteinase expression, and possible genetic alterations such as ICA1L variants are also implicated. **Conclusions**: This case underscores the importance of considering molecular mechanisms in the pathophysiology of delayed post-endovascular treatment of ICH, as well as the need for hemodynamic monitoring and follow-up in patients with vascular comorbidities.

## 1. Introduction and Clinical Significance

Carotid-cavernous fistulas (CCFs) are abnormal arteriovenous communications, typically acquired and most often secondary to head trauma [1]. They consist of a high-flow arterial shunt from the cavernous segment of the internal carotid artery into the cerebral venous system, leading to orbitocranial venous hypertension, visual disturbances, and, in some cases, distal arterial hypoperfusion [2]. Endovascular treatment has proven to be safe and effective, although abrupt interruption can be associated with ischemic and hemorrhagic complications [3].

Intracerebral hemorrhage (ICH) is the second-most common form of stroke and affects around 2 million people worldwide annually and carries a high mortality rate, reaching up to 34.7% at 30 days and 45.1% at one year. In survivors, ICH is also associated with significant disability [4,5,6,7]. This underscores the need to understand the factors involved in its pathophysiology and management [8].

There are several causes of ICH; the most common are summarized in Table 1 [9,10,11,12,13]. ICH caused by the closure of a CCF is very rare. Some pathophysiological mechanisms have been postulated to explain this hemorrhagic complication, such as venous thrombosis in the affected venous bed or incomplete closure of the CCF that eventually leads to rupture of the veins. To our knowledge, there are only two reported cases of ICH after complete CCF closure due to arterial bleeding, not associated with another vascular malformation [14,15], as presented in this case (Table 2).

The most widely accepted theory to explain the pathophysiology of arterial hemorrhage after a complete CCF closure is Spetzler’s “Normal Perfusion Pressure Breakthrough” (NPPB), which suggests that distal vasoregulatory dysfunction occurs as a consequence of sustained hypoperfusion [16].

We describe the case of a traumatic CCF who developed intracranial hemorrhage 12 days after endovascular closure with coils. To our knowledge, this is the only case in the medical literature with such a prolonged interval between CCF closure and arterial hemorrhagic complication.

The aim of this report is to illustrate a particular case and discuss possible pathophysiological and molecular mechanisms, considering arterial hypertension as a risk factor. From this integrative perspective, we seek to gather information on the pathophysiological events that may underlie delayed ICH and to guide future follow-up and management strategies.

We report a rare case presenting to the emergency room with acute intracerebral hemorrhage twelve days after successful CCF closure.

A comprehensive literature review was conducted using the PubMed database, covering publications from 1990 to 2025, in both English and Spanish. The search strategy focused on identifying all articles reporting cases of cerebral hemorrhage following the closure of a carotid-cavernous fistula. Only studies that described hemorrhagic events not associated with other cerebral vascular abnormalities were included. This approach ensured the selection of relevant literature for the analysis of this specific complication.

## 2. Case Presentation

A 48-year-old woman with a history of arterial hypertension, chronic hepatitis B virus infection treated with tenofovir, and anxiety. She was admitted to the emergency department because of a high-energy traffic accident. On initial evaluation, a traumatic rupture of the sigmoid colon was identified, which was surgically treated with a Hartmann procedure, and she was admitted to the general surgery department for postoperative management.

Two days after admission, the patient developed right conjunctival injection, ophthalmoparesis, and decreased visual acuity. An ophthalmological assessment was conducted, consisting of a fundus examination, ocular tonometry, and optical coherence tomography, all with normal results. Brain magnetic resonance imaging (MRI) showed right orbital edema and punctate hyperintense lesions in the right frontal lobe. These findings were initially interpreted as post-traumatic changes not requiring specific intervention. The patient was discharged home after a few days.

Two months after the accident, the patient’s visual symptoms worsened, leading to loss of vision in the right eye. She was re-evaluated by an ophthalmologist, and brain MR angiography (MRA) was performed, which revealed a high-flow direct right carotid-cavernous fistula, type A according to the Barrow classification. The patient underwent diagnostic arteriography that confirmed the diagnosis and showed significant steal of blood flow from the right internal carotid artery, venous congestion, and flow reversal in the Sylvian vein and the right ophthalmic vein. The cavernous sinus was observed to be greatly enlarged, indicating high pressure through the CCF. It should be noted that perfusion of the right cerebral hemisphere was mainly supplied by the vertebrobasilar system through the right posterior communicating artery and by the left carotid axis through the anterior communicating artery. These findings demonstrate hypoperfusion of the right cerebral hemisphere.

The day after the diagnosis, our patient underwent therapeutic arteriography and was successfully treated with transvenous coil embolization, achieving complete closure of the CCF without complications. She was discharged after 24 h.

Twelve days after CCF closure, the patient was admitted to the emergency department with an acute neurological condition characterized by disorientation in time and space, severe headache, neck pain, bradypsychia, apathy, anorexia and asthenia. On physical examination, left facial paralysis was noted, in addition to the previously known right-eye blindness (amaurosis). A cranial CTA showed an intracerebral hemorrhage in the right frontal lobe (Figure 1 and Figure 2), the same region previously affected by hypoperfusion due to the CCF.

Since MR venography and a new cerebral arteriography were normal, fistulous recanalization, cerebral venous thrombosis, or residual vascular malformations were ruled out. A few days later, the patient’s neurological condition worsened: she became drowsy, did not follow complex commands, and developed severe hemiparesis of the left arm and leg. A cranial CTA revealed significant perilesional edema causing midline shift in the brain. She was evaluated by neurosurgery, but surgical intervention was not performed; instead, furosemide and dexamethasone were added to her treatment. Within a day, the patient showed clear improvement, with only mild left hemiparesis remaining, and was subsequently discharged. Three months later, the patient still had amaurosis of the right eye and left facial paresis, but no other neurological symptoms.

The radiological progression of this clinical case is summarized in Figure 1.

## 3. Discussion

A narrative literature review was conducted using three major databases: PubMed, Scopus, and the Cochrane Library. The search aimed to identify studies describing intracerebral hemorrhage as a direct consequence of carotid-cavernous fistula closure. Only articles written in English and involving adult patients were considered eligible for inclusion. The search strategy employed the following terms: “Arteriovenous malformation,” “Perfusion pressure,” “Perfusion pressure breakthrough,” “Occlusive hyperemia,” and “Hemorrhage.” This methodological approach allowed for a comprehensive synthesis of the existing evidence on this rare clinical outcome.

Intracerebral hemorrhages (ICH) are a devastating type of stroke, with high morbidity and mortality and persistent neurological sequelae [10,17,18,19]. Although chronic arterial hypertension and cerebral amyloid angiopathy are the main etiological factors, multiple causes have been identified (Table 1) [13,20,21]. It is worth noting that, according to our center’s registry, there has not been a single case with an etiology similar to the one presented in this report over the past 24 years.

The Stroke Registry of Hospital Sagrat Cor in Barcelona (*n* = 3741, over 24 years), shows that ICH is associated with higher in-hospital mortality (27.1% vs. 12.6%), more infectious, neurological, and systemic complications, and worse functional recovery than ischemic stroke. These data are consistent with the Mataró Registry and other regional studies [22,23].

We present the case of a cerebral hemorrhage following endovascular closure of a direct carotid-cavernous fistula (CCF), a rare entity that invites reflection from a clinical, pathophysiological, and molecular perspective.

Cerebral hemorrhage is a rare complication, typically observed within the first 48 h after treatment of a CCF [14,15]. The first published case was in 2002, Cloft et al. reported a case of a 74-year-old woman who, two days after balloon closure, developed right hemiparesis and underwent a CTA that showed hemorrhage in the left basal ganglia, resulting in the patient’s death [14]. In 2011, Cho et al. [15] published the last known case, a 48-year-old male with bilateral traumatic CCF who presented with a right parietal hemorrhage on the control CTA one day later. In both cases, the hemorrhage occurred in the same hypoperfused arterial bed due to diversion of arterial blood flow toward the venous circulation. In both cases, no cerebral venous thrombosis or malformations, such as arterial aneurysms, were found to explain the hemorrhage. The most significant feature shared by these cases is that the cerebral hemorrhage occurred shortly after complete CCF closure. The interval between CCF closure and hemorrhage was approximately 24 h. The characteristics of these hemorrhages are highly suggestive of being caused by arterial reperfusion at normal pressure in an arterial bed that had been subjected to prolonged hypoperfusion. Thus, normal arterial pressure becomes excessive for a hypoperfused arterial bed, resulting in the rupture of these arteries. This is supported by the Normal Perfusion Pressure Breakthrough (NPPB) theory. The main characteristics of the two previously reported cases, compared with ours, are summarized in Table 2.

In 1978, Spetzler et al. described the malignant edema or hemorrhage that sometimes occurs in the ipsilateral hemisphere of a high-flow arteriovenous malformation (AVM) following resection. They coined the term “normal perfusion pressure breakthrough (NPPB)” [16] to explain the rupture of arteries under normal perfusion pressures. To explain this, it is essential to understand the cerebral pressure autoregulation, which is the brain’s intrinsic capacity to maintain stable cerebral blood flow (CBF) across a wide range of systemic blood pressures. The mechanisms governing cerebral perfusion pressure (CPP) autoregulation safeguard neural tissue by preventing cerebral ischemia during episodes of hypotension and by mitigating the risk of hyperemia, which may lead to capillary damage, edema, or hemorrhage, during periods of hypertension [24].

In NPPB theory, normal vessels adjacent to arteriovenous malformations (AVMs) remain chronically dilated to preserve cerebral blood flow, potentially impairing autoregulatory capacity. Post-resection, redirected flow into these low-resistance vessels overwhelms compromised autoregulation (typically regulated at arteriolar levels), leading to capillary damage, edema, or hemorrhage—consistent with Normal Perfusion Pressure Breakthrough (NPPB). Supporting evidence includes angiography and transcranial Doppler studies demonstrating impaired autoregulation in AVM-feeding vessels, microvascular flow increases post-resection, and in vitro nonreactivity of vessels from patients with post-resection complications. However, contradictory data show restored CO_2_ reactivity post-resection, intact vasoreactivity in NPPB cases, and preserved pressure autoregulation despite improved perfusion, challenging the universality of NPPB [25,26,27,28,29,30,31,32,33].

In the context of arteriovenous malformations (AVMs), the diversion of arterial circulation into the venous system can lead to hypotension within the arterial cerebral circulation, likely due to the substantial volume of blood shunted through low-resistance vessels, which diverts flow from the parallel normal vasculature. Studies have demonstrated that the mean arterial pressures (MAPs) in arteries supplying AVMs are reduced by approximately 50% compared to systemic MAPs. Measurements obtained during superselective cerebral angiography indicate a progressive decline in intra-arterial pressure as one moves distally along the arterial tree. Furthermore, evidence suggests that regions of normal brain tissue in the hemisphere ipsilateral to the AVM receive blood from arteries exhibiting marked relative hypotension. Collectively, these findings suggest that an AVM functions as a “buffering” system, such that fluctuations in systemic MAP are not effectively transmitted to the vasculature in close proximity to the AVM [32,34,35,36].

More recent research found that in normal brain tissue near AVMs, raising systemic blood pressure did not increase regional blood flow in low-pressure territories. This means that long-term low blood pressure does not always damage the brain’s ability to regulate blood flow. Instead, the point at which this regulation starts is lower. Also, in AVM models, new capillaries often form without full support from surrounding astrocytes, making them weaker and less stable [35,37,38,39].

In our case, the delayed onset of ICH, twelve days after CCF closure, suggests that pathophysiological changes in NPPB theory remain for several days; this requires consideration of other possible concurrent pathophysiological processes, including those of an inflammatory basis.

### Molecular Basis of Normal Perfusion Pressure Breakthrough Theory

Under physiological conditions, vascular smooth muscle cells respond through adaptive contraction or dilation, modulated by intracellular calcium signals, transient receptor potential (TRP) ion channels [40], 20-Hydroxyeicosatetraenoic acid (20-HETE) [41], prostanoids, and nitric oxide [42]. In this context, endothelial dysfunction in the setting of conditions such as arterial hypertension contributes to increased vulnerability, favoring rupture in situations of hyperperfusion or acute hemodynamic stress [43,44,45,46].

At the molecular level, chronic vasodilation is mediated by upregulation of vasodilatory pathways, including increased production of nitric oxide (NO) and prostaglandins, and downregulation of vasoconstrictive responses. Over time, the smooth muscle cells in the vessel wall lose their contractile responsiveness, a phenomenon termed “vasomotor paralysis” [27,47]. This is accompanied by structural changes such as increased capillary density, thinning of vessel walls, and loss of astrocytic foot processes, which compromise the integrity of the blood–brain barrier (BBB) [37].

When the AVM is resected, normal perfusion pressure is suddenly restored to these chronically dilated, autoregulation-deficient vessels. The inability of these vessels to constrict in response to increased pressure results in hyperperfusion, increased transcapillary hydrostatic pressure, and subsequent BBB disruption. This leads to extravasation of plasma proteins, cerebral edema, and, in severe cases, parenchymal hemorrhage. Experimental models confirm that BBB breakdown and neuronal injury are associated with this loss of autoregulatory capacity and can be mitigated by agents that restore vascular tone, such as indomethacin, which inhibits prostaglandin synthesis [24,37,47,48,49,50].

Currently, multiple animal models have been developed to replicate this phenomenon. In a well-established rat model, chronic cerebral hypoperfusion is induced by creating arteriovenous fistulas (AVFs), leading to sustained vasodilation and impaired autoregulation. Upon occlusion of the AVF (mimicking AVM resection), there is a rapid increase in perfusion pressure, resulting in increased intracranial pressure, BBB disruption (as evidenced by sodium fluorescein extravasation), and histological evidence of diffuse ischemic changes and neuronal apoptosis—findings consistent with NPPB. Pharmacologic intervention with indomethacin in this model partially reverses these hemodynamic and histological changes, further supporting the role of impaired autoregulation in the pathophysiology of NPPB [50].

In feline models, chronic ischemia is induced by creating carotid-jugular fistulas and vertebral artery occlusion. Restoration of normal flow or induction of hypertension in these animals leads to persistent pial arterial dilation, extensive BBB disruption, cerebral edema, and infarction, particularly after fistula occlusion. Control animals with intact autoregulation do not develop these changes, highlighting the critical role of autoregulatory failure in NPPB [49].

Structural studies in rats with chronic AVFs demonstrate increased capillary density and loss of astrocytic foot processes, indicating neovascularization and mechanical vulnerability of the microvasculature, which predisposes to edema and hemorrhage upon restoration of normal perfusion pressure [37].

Collectively, these animal experiments provide robust mechanistic evidence for the NPPB theory, demonstrating that chronic hypoperfusion impairs autoregulation and microvascular integrity, and that restoration of normal perfusion pressure precipitates hyperperfusion injury, BBB breakdown, and parenchymal damage [47].

On the other hand, cerebral hyperperfusion syndrome following severe carotid stenosis reperfusion is a situation quite similar to the pathophysiological scenario of this case report. In both situations, there is a hypoperfused cerebral area that rapidly regains cerebral blood flow. Thus, some molecular pathophysiological characteristics could be shared by both conditions. At the molecular level, reperfusion triggers a burst of reactive oxygen species (ROS), which damage cellular membranes, proteins, and DNA. ROS also activate matrix metalloproteinases (MMPs), particularly MMP-9, which degrade tight junction proteins and further disrupt the BBB. This facilitates extravasation of plasma proteins and leukocytes, promoting vasogenic edema and, in severe cases, intracerebral hemorrhage.

Additionally, reperfusion activates inflammatory pathways, microglia and infiltrating leukocytes release cytokines (e.g., TNF-α, IL-1β), amplifying endothelial injury and neuronal apoptosis. Mitochondrial dysfunction and calcium overload in neurons and glia further contribute to cell death [51]. As we have seen, inflammation plays an important role. It should be noted, on the other hand, that some experiments have shown that substances such as Triggering Receptor Expressed on Myeloid Cells 2 (TREM2) attenuate neuroinflammation and apoptosis [52].

The neurovascular unit (endothelial cells, astrocytes, pericytes, neurons) is central to these processes, with crosstalk between its components mediating both the acute injury and subsequent repair mechanisms. The net result is a spectrum of clinical manifestations, from headache and seizures to devastating hemorrhagic transformation [53,54,55].

The pathophysiology described in the medical literature originates from foundational animal studies. These studies have shown that after prolonged carotid occlusion, reperfusion results in excessive cerebral blood flow, blood–brain barrier disruption, vasogenic edema, and sometimes hemorrhage, due to the inability of previously dilated arterioles to constrict in response to restored perfusion pressure. The American Heart Association, in its guidelines, explicitly references this mechanism as analogous to the “normal perfusion pressure breakthrough” phenomenon observed in animal models of arteriovenous malformation resection, further supporting the translational relevance of animal data to human CHS.

Although the references do not cite specific animal studies by author or year, the described mechanisms and clinical observations are directly extrapolated from such experimental work, which has established the loss of autoregulation and reperfusion injury as central to CHS pathogenesis [53,56,57].

Individual susceptibility to ICH could in turn be modulated by genetic variants involved in vascular integrity, inflammation, or small vessel disease. Recent studies identify epigenetic alterations and genes such as ICA1L related to wall integrity and the stability of endothelial junctions, which themselves are associated with the risk of ICH and progression of vascular damage [45,58,59].

In summary, this case highlights the importance of close follow-up after endovascular treatment of a CCF, especially in patients with hypertension or other vascular risk factors. From a molecular perspective, it is essential to advance the identification of inflammatory, angiogenic, and genetic biomarkers to stratify the risk of post-intervention ICH.

The integration of these findings into clinical practice could be relevant for the prevention and management of hemorrhagic complications in high-risk patients [18,60,61].

## 4. Conclusions

Late intracerebral hemorrhage following closure of a direct carotid-cavernous fistula is an uncommon complication and has not previously been reported with a latency greater than 48 h. We report a hemorrhagic complication twelve days after successful coil treatment in a patient with a history of arterial hypertension and no residual vascular malformation.

This case is notable for the delayed occurrence of intracranial hemorrhage—twelve days after complete closure of a cranial arteriovenous fistula—occurring in the setting of chronic arterial hypertension, and it suggests the involvement of broader pathophysiological mechanisms beyond those classically proposed by the Normal Perfusion Pressure Breakthrough (NPPB) theory.

The review of this case, together with the literature, points to the influence of cerebral autoregulation dysfunction along with molecular mechanisms such as microglial activation, and overexpression of metalloproteinases, among others, in conditioning greater capillary susceptibility to bleeding. In addition, possible genetic factors are highlighted, such as variants in ICA1L and epigenetic profiles associated with a higher risk of vascular rupture.

Based on our review, clinical management protocols for patients undergoing closure of carotid-cavernous fistulas should prioritize stringent blood pressure control, particularly in individuals with a history of hypertension. We recommend maintaining blood pressure strictly within normal or even low-normal ranges to minimize the risk of hemorrhagic complications following fistula closure. Furthermore, our findings suggest that a staged or gradual closure approach may offer additional benefits, as it could allow the cerebral arterial system to progressively adapt to restored hemodynamics and regain appropriate vasoreactivity, thereby reducing the likelihood of reperfusion injury.

In summary, this report emphasizes the need for blood pressure control in the management of a CCF and the importance of ensuring individualized clinical follow-up in patients with vascular risk factors. A clinical–molecular analysis supports future research into biomarkers that allow for risk stratification and guide therapeutic strategies in this type of postoperative complication.

## Figures and Tables

**Figure 1 reports-08-00234-f001:**
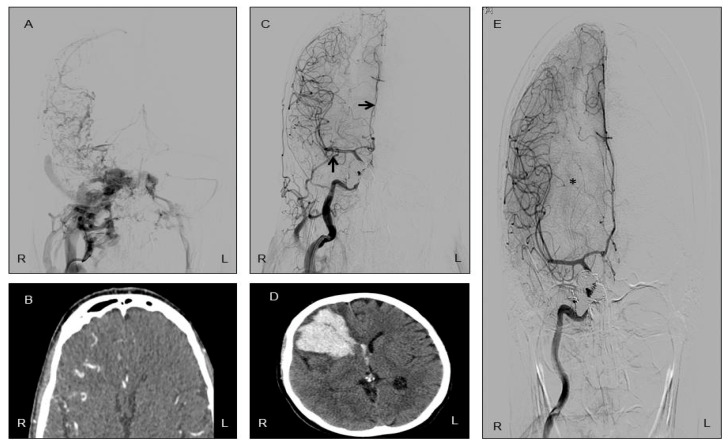
Radiological evolution of intracerebral hemorrhage following successful closure of a carotid-cavernous fistula (CCF). (**A**) Carotid arteriography showing a right-sided CCF with bilateral venous drainage. Note the absence of contrast opacification in the right middle cerebral artery (MCA) and right anterior cerebral artery (ACA), indicating impaired arterial flow prior to treatment. (**B**) CT angiography (CTA) revealing hyperdense and tortuous venous structures in the right hemisphere, consistent with abnormal venous drainage secondary to the ipsilateral CCF. (**C**) Immediate post-procedural arteriography acquired after endovascular occlusion of the CCF; arrows highlight the reappearance of the right ACA and MCA, suggesting reestablishment of normal arterial circulation. (**D**) CTA performed twelve days after CCF closure demonstrates a right frontal lobe intracerebral hemorrhage, reflecting delayed vascular reperfusion injury. (**E**) Follow-up arteriography one month later shows stable vessel patency and resolution of venous congestion. The asterisk identifies the residual mass effect produced by the hematoma. (R: right, L: left).

**Figure 2 reports-08-00234-f002:**
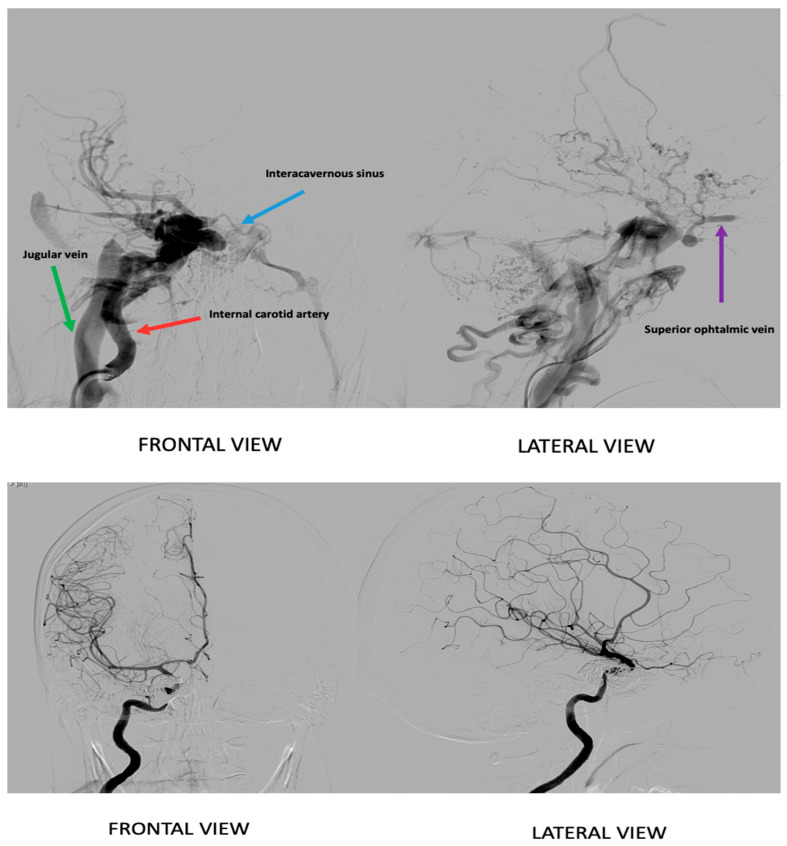
Selective angiography from: At the **top**, an angiographic image depicts the carotid-cavernous fistula prior to treatment. The **lower** image shows the angiography after treatment, demonstrating the post-intervention vascular status.

**Table 1 reports-08-00234-t001:** Causes of acute spontaneous intracerebral hemorrhage.

Arterial Hypertension (Deep Perforating Vasculopathy)	Cerebral Amyloid Angiopathy
Acute Arterial hypertension - Drugs - Cold exposure - Trigeminal nerve stimulation - Cardiac catheterization - Insect bites (wasp, viper) - Electroshock	Cerebral vascular malformations - Cerebral arteriovenous malformation - Cavernous malformation - Venous angioma - Dural arteriovenous fistula - Intracranial arterial aneurysm
Intracranial venous thrombosis (CVT) - CVT from local cause (head trauma, head and CNS infections, tumor) - CVT from systemic cause (pregnancy and puerperium, cancer, prothrombotic disorder)	Hemorrhagic transformation of cerebral infarction - Spontaneous hemorrhagic infarction - Hemorrhagic infarction occurring in patients receiving anticoagulant or thrombolytic therapy
Cerebral tumors - Primary tumor - Metastatic (mostly from melanoma, kidney and lung)	Hemostatic and hematologic disorders - Primary coagulopathy - Severe thrombocytopenia - Severe clotting factor deficiency such as hemophilia - Afibrinogenemia - Secondary coagulopathy - Antithrombotic drugs
Vasculitis and related vasculopathies - Systemic vasculitis - Isolated Primary angiitis of the central nervous system - Reversible cerebral vasoconstriction syndrome - Posterior reversible encephalopathy syndrome - Infective endocarditis - Infections of the CNS - Other vasculopathies	Changes in cerebral blood flow - Cardiac surgery - Carotid endarterectomy or thrombectomy
Toxic - Cocaine - Other sympathomimetic drugs Rare entities - Dissection of intracranial arteries - Hyperperfusion syndrome Other - Migraine - Alcohol abuse - Familial CAA - CADASIL - COL4A1 mutations	

Modified from Mendiola et al. [10]

**Table 2 reports-08-00234-t002:** Reported cases of intracerebral hemorrhage complicating the closure of a carotid-cavernous fistula.

Case Features	Case 1	Case 2	Case 3
Author	Cloft et al. [14]	Cho et al. [15]	Present Case
Year of publication	2002	2011	2025
Age	74 years	48 years	48 years
Sex	Female	Male	Female
Interval from CCF closure to ICH	40 h	24 h	12 days
Side of hemorrhage	Left basal ganglia	Right parietal lobe	Right frontal lobe
Result	Deceased	Recovery	Recovery

## Data Availability

The data that support the findings of this study are available on request from the corresponding author. The data are not publicly available due to privacy or ethical restrictions.

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
