# Peer review of "Late Intracerebral Hemorrhage After Successful Endovascular Closure of a Carotid-Cavernous Fistula: A Case Report and Updated Review"

_reports, 2025, doi:10.3390/reports8040234_

Round 1

Reviewer 1 Report

Comments and Suggestions for Authors

This report is highly informative. By detailing the unique case of late ICH (12 days) after successful CCF closure, the authors successfully argue for an expanded understanding of post-reperfusion injury that includes molecular and inflammatory mechanisms alongside the classic NPPB theory. The manuscript is well-written, thoroughly referenced, and provides valuable clinical guidance regarding post-procedural monitoring, especially for patients with vascular risk factors like hypertension. I strongly recommend this paper for publication.

Minor Comments

The literature search methodology could be detailed more explicitly (e.g., databases, search terms, inclusion/exclusion criteria).

Figure legends would benefit from greater clarity and descriptive detail for readers unfamiliar with this pathology.

Minor typographical and formatting edits are needed to improve readability and consistency.

Consider expanding on the implications for clinical management protocols or follow-up standards.

Author Response

This report is highly informative. By detailing the unique case of late ICH (12 days) after
successful CCF closure, the authors successfully argue for an expanded understanding of post-
reperfusion injury that includes molecular and inflammatory mechanisms alongside the classic
NPPB theory. The manuscript is well-written, thoroughly referenced, and provides valuable
clinical guidance regarding post-procedural monitoring, especially for patients with vascular risk
factors like hypertension. I strongly recommend this paper for publication.
Minor Comments
The literature search methodology could be detailed more explicitly (e.g., databases, search
terms, inclusion/exclusion criteria).

We sincerely appreciate the reviewer’s interest in the methodological details of
our narrative review.
A comprehensive search was conducted across three electronic databases:
PubMed, Scopus, and Cochrane. We restricted our selection to articles explicitly
reporting intracerebral hemorrhage as a direct complication following the
closure of a carotid-cavernous fistula. Only articles published in English and
pertaining to adult patients were considered.
For the literature search, the following terms were employed: “Arteriovenous
malformation,” “Perfusion pressure,” “Perfusion pressure breakthrough,”
“Occlusive hyperemia,” and “Hemorrhage.”
To enhance transparency, we have now included a dedicated paragraph in the
Methods section describing this approach in greater detail, as recommended.

Figure legends would benefit from greater clarity and descriptive detail for readers unfamiliar
with this pathology.We sincerely thank the reviewer for this helpful suggestion. We have revised
the figure legends and added further descriptive details to ensure that the
figures are now clearer and more easily interpretable for a broad readership.
These improvements aim to enhance the overall comprehensibility and visual
coherence of the illustrations within the manuscript.

Minor typographical and formatting edits are needed to improve readability and consistency.
We sincerely appreciate the reviewer’s attentive observation. All typographical
and formatting inconsistencies have been thoroughly revised to ensure
improved readability, uniform style, and overall consistency throughout the
manuscript. We have carefully reviewed headings, figure legends, references,
and text alignment to guarantee a polished and professional presentation.

Consider expanding on the implications for clinical management protocols or follow-up
standards.
We appreciate the reviewer’s thoughtful suggestion. We have expanded the
Discussion section to further address the clinical implications of our findings.
Specifically, we now elaborate on how this case may inform current
management protocols and follow-up strategies for patients undergoing CCF
closure. The revised text highlights the importance of close post-procedural
monitoring and individualized hemodynamic assessment to prevent delayed
complications, thereby providing clearer guidance for future clinical practice.

Reviewer 2 Report

Comments and Suggestions for Authors
  • What is the main question addressed by the research?

Does a delayed intracerebral hemorrhage (ICH) occurring 12 days after successful endovascular closure of a direct carotid-cavernous fistula (CCF) occur, and what pathophysiology (beyond classic NPPB) could explain it, and what are the monitoring implications?

  • Do you consider the topic original or relevant to the field? Does it address a specific gap in the field? Please also explain why this is/ is not the case.

Relevant to neurointerventional complications; modest originality. Prior reports describe ICH within 24–48 h; this case extends the window to 12 days, which is unusual and clinically useful (follow-up/monitoring). However, it remains a single case, and the mechanistic section is largely narrative/speculative, lacking patient-level biomarkers. The gap is partially addressed: the paper flags a delayed risk period but does not provide new mechanistic evidence.

  • What does it add to the subject area compared with other published material?

Adds a well-documented, latest-timed case and a compact comparison to two earlier cases (Table 2), plus a synthesized discussion of NPPB and ancillary molecular hypotheses (oxidative stress, BBB/MMPs, microglia, genetic susceptibility).

  • What specific improvements should the authors consider regarding the methodology?

Need the IRB/consent statement and get the IRB approval.

  • Are the conclusions consistent with the evidence and arguments presented and do they address the main question posed? Please also explain why this is/is not the case.

Yes. The mechanistic expansion beyond NPPB (ROS/MMPs/microglia/genetics) is plausible but hypothesis-level in this report (no direct biomarker/imaging evidence provided), so the tone should remain cautious.

  • Are the references appropriate?

OK.

  • Any additional comments on the tables and figures.

Table 3 (local registry ICH vs ischemic stroke outcomes): not necessary for this case.

  1. Table 3 isn’t necessary for this case report.
  2. Please give the IRB number.
  3. Case 3 in Table 2 of the author may change to “Present case”.
  4. Please tighten the Discussion section for the case report.

Author Response

• What is the main question addressed by the research?
Does a delayed intracerebral hemorrhage (ICH) occurring 12 days after successful
endovascular closure of a direct carotid-cavernous fistula (CCF) occur, and what
pathophysiology (beyond classic NPPB) could explain it, and what are the monitoring
implications?
We appreciate the reviewer’s insightful summary of the manuscript’s core
objectives. Indeed, our study was structured with these guiding questions at the
forefront. In response, the manuscript first emphasizes the clinical relevance and
necessity of reporting such cases to raise awareness and contribute to evidence-
based practice. We then provide a thorough literature review to contextualize
the frequency of similar cases. The physiopathological mechanisms underlying
the complication are critically analyzed and discussed. Subsequently, preventive
measures are suggested based on both observed outcomes and existing
guidelines. Finally, the conclusion includes recommendations for future research
priorities, with the intent to foster continued advancement in understanding and
mitigating this specific complication. These points have been clarified and
highlighted throughout the revised manuscript to ensure their prominence and
coherence.

• Do you consider the topic original or relevant to the field? Does it address a specific gap
in the field? Please also explain why this is/ is not the case.
Relevant to neurointerventional complications; modest originality. Prior reports describe ICH
within 24–48 h; this case extends the window to 12 days, which is unusual and clinically useful
(follow-up/monitoring). However, it remains a single case, and the mechanistic section is largely
narrative/speculative, lacking patient-level biomarkers. The gap is partially addressed: the paper
flags a delayed risk period but does not provide new mechanistic evidence.
We thank the reviewer for this pertinent question. To the best of our
knowledge, this type of complication has been reported in the literature only
twice previously, making our report the third documented case. The
complication described is severe, and it is plausible that additional cases
remain unpublished due to various factors. Thus, we consider it highly relevant
for the field to recognize the existence of this complication and understand its
underlying mechanisms. Our publication explicitly identifies several significant
gaps in current knowledge. Notably, the frequency of this complication remains
undefined due to scarcity of reports. Furthermore, its physiopathology warrants
elucidation, as prior investigations have predominantly focused on
complications associated with arteriovenous malformations rather than carotid-
cavernous fistulas, which differentiate the present case. Lastly, a major
knowledge gap exists regarding effective preventive measures for this
complication. We believe addressing these gaps reinforces the originality and
clinical importance of our work.
Currently, the evidence base remains extremely limited, restricting our ability to
delve deeply into alternative mechanistic pathways beyond those already
mentioned. The paucity of patient-level biomarker data and the rarity of
reported cases render extensive mechanistic elucidation difficult at this stage.
Our case primarily aims to highlight the extended risk window and to serve as a
catalyst for further fundamental research in basic sciences. Such investigations
are crucial to bridge these significant gaps in understanding and to develop
mechanistically grounded preventive and therapeutic strategies. Thus, while
mechanistic evidence remains preliminary, this report underscores the urgent
need for dedicated basic science studies to resolve these knowledge deficits.

• What does it add to the subject area compared with other published material?
Adds a well-documented, latest-timed case and a compact comparison to two earlier cases
(Table 2), plus a synthesized discussion of NPPB and ancillary molecular hypotheses (oxidative
stress, BBB/MMPs, microglia, genetic susceptibility).
We fully agree with the reviewer’s assessment and appreciate the
acknowledgment of the comprehensive documentation and comparative
synthesis presented in our manuscript.

• What specific improvements should the authors consider regarding the methodology?
Need the IRB/consent statement and get the IRB approval.
We acknowledge this requirement and will ensure that the Institutional Review
Board approval and consent statement are appropriately obtained and
included in the revised manuscript as per journal and ethical standards

• Are the conclusions consistent with the evidence and arguments presented and do they
address the main question posed? Please also explain why this is/is not the case.
Yes. The mechanistic expansion beyond NPPB (ROS/MMPs/microglia/genetics) is plausible but
hypothesis-level in this report (no direct biomarker/imaging evidence provided), so the tone
should remain cautious.
We appreciate this observation and will ensure that the discussion maintains
an appropriately cautious tone, explicitly acknowledging the hypothesis-level
nature of the mechanistic proposals in the absence of direct biomarker or
imaging evidence
• Are the references appropriate?
OK.

Round 2

Reviewer 1 Report

Comments and Suggestions for Authors

The manuscript has been sufficiently improved to warrant publication in Reports. 

Reviewer 2 Report

Comments and Suggestions for Authors

The authors had answered my questions.